# High Thermal Conductivity and Anisotropy Values of Aligned Graphite Flakes/Copper Foil Composites

**DOI:** 10.3390/ma13010046

**Published:** 2019-12-20

**Authors:** Fankun Zeng, Chen Xue, Hongbing Ma, Cheng-Te Lin, Jinhong Yu, Nan Jiang

**Affiliations:** 1Key Laboratory of Marine Materials and Related Technologies, Zhejiang Key Laboratory of Marine Materials and Protective Technologies, Ningbo Institute of Materials Technology and Engineering (NIMTE), Chinese Academy of Sciences, Ningbo 315201, China; zengfankun@nimte.ac.cn (F.Z.); mahongbing@nimte.ac.cn (H.M.);; 2Center of Materials Science and Optoelectronics Engineering, University of Chinese Academy of Sciences, Beijing 100049, China

**Keywords:** alignment of graphite flakes, copper, thermal conductivity, hot press process

## Abstract

Much attention has been paid to graphite flakes/copper (GFs/Cu) composites for thermal management due to their remarkable thermal properties. Most studies focus on the interface interaction between GFs and Cu in composites. However, controlling the orientation of GFs still remains a challenge. Herein, we report a reliable method to ensure consistent orientation of GFs in the composites. Firstly, the disorder GFs were well arranged on the surface of copper foil by tape casting process in the casting machine. Then highly aligned GFs/Cu composites were fabricated by hot pressing process in a vacuum hot-pressing furnace, with the volume fraction of graphite from 30% to 70%. The SEM images show that the obtained GFs/Cu composites presented a layer-by-layer structure or network structure with a different content of GFs. The thermal conductivity of GFs/Cu composites exhibited an extreme anisotropy due to the highly aligned GFs. The ultrahigh thermal conductivity of GFs/Cu composites with 70 vol% GFs reached 741 W/(m·K), while through-plane thermal conductivity was just 42 W/(m·K). The alignment of GFs and interfacial thermal resistance were deeply analyzed and a thermal conductivity model for GFs/Cu composites was established. Our work provides a new idea to significantly enhance the thermal transportation performance of GFs/Cu composites by well controlled alignment of GFs in Cu matrix.

## 1. Introduction

In recent years, heat sink materials with high thermal conductivity (TC) have become increasingly important, especially in packaging electronic chips and devices with high power density which produce much heat to dissipate [1,2]. Owing to good thermal and mechanical properties, Al with a TC of 237 W/(m·K) and Cu with a TC of 400 W/(m·K) are frequently used as the raw materials of heat sinks and heat spreaders in order to transfer heat from electronic devices to outside equipment [3,4]. However, the heat dissipation of Al cannot match the requirements of modern power electronics and Cu is too heavy to be used in integrated and miniaturized electronic devices. Carbon-based materials (e.g., graphene [5,6], graphite flakes [7,8], carbon nanotubes [9], carbon fibers [10], and diamond [11], etc.) are promising conductive fills due to their high TC. The 1D carbon nanotubes, which are typical nano carbon materials, have quoted values of ~3500 W/(m·K) with single-walled structures and ~3000 W/(m·K) with multi-walled structures in axial directions at room temperature [12,13]. The TC of carbon fibers reaches 1000 W/(m·K) in the longitudinal direction [14], which is lower than carbon nanotubes due to the increase in size. The 2D graphene exhibits a high TC in the range of 1500~5300 W/(m·K) at room temperature, depending on the defects and thickness [15,16]. The in-plane ultimate maximum TC of graphite is approximately 2000 W/(m·K) [17,18]. The 3D diamond, the hardest matrials in nature, are considered perfect thremal and mechanical reinforcements with TC in a range from 1200 to 2000 W/(m·K) [19,20].

Nowadays, carbon-based materials to reinforce metal matrix composites show promise for thermal management. However, the distribution and alignment of nanoscale carbon-based materials (e.g., graphene and carbon nanotubes, etc.) remain big challenge to researchers, limiting the content of conductive fills in composites [21]. Diamond/metal matrix composites draw much attention due to excellent thermal properties, with isotropic TC approaching 600 W/(m·K) [22]. Nevertheless, the high price of diamonds and the lack of easy processability precludes the using of these composites from a wide range of technological applications. In view of this, graphite flakes (GFs)/Al and GFs/Cu composites have attracted much interest from researchers, with high TC, low cost, good mechanical properties and processability. It is reported that interface bonding significantly affects the TC and mechanical properties of GFs/metal composites, due to bad wettability of GFs and metals. Xue et al. [8] reported that a silicon carbide nano-layer coated on the GFs’ surface could increase interface bonding of GFs/Al composites. The in-plane TC and flexural strength of GFs/Al composites increased from 505~702 W/(m·K), 46~24 MPa to 528~735 W/(m·K), 90~45 MPa, respectively, as the volume fraction of GFs increased from 40% to 70%. Bai et al. [23] reported that a boron carbide nano-layer on the graphite surface realized by salt bath process could promote flexural strength of GFs/Cu composites from 94~52 MPa to 145~74 MPa, as the volume fraction of GFs was 40% to 70%. Howerer, the in-plane TC of GFs/Cu composites decreased from 477~676 W/(m·K) to 449~608 W/(m·K). A silicon carbide nano-layer on GFs surface could also enhance interface bonding of GFs/Cu composites, with flexural strength increasing from 94~52 MPa to 110~75 MPa [24]. Nevertheless, the in-plane TC decreased from 474~676 W/(m·K) to 460~610 W/(m·K), similar to carbide-boron nano-layer coated GFs/Cu composites. Although coatings of the boron carbide nano-layer and silicon carbide nano-layer were of benefit to interface bonding of GFs/Cu composites, they may not be of benefit to TC. Liu et al. [25] fabricated 71 vol% of GFs/Cu composites using a sparking plasma sintering after Cu coated on GFs by electroless plating process and improved TC to 565 W/(m·K). However, the value of TC was too low. Some researchers have disscussed the effect of shape or average size, volume fraction, and alignment of GFs on the TC of GFs/metal composites. Sohn et al. [26] reported that the TC of GFs/Cu composites fabricated by hot-pressing method reached 456.9 W/(m·K) with 59 vol% of GFs, which was approximately twice that of granules/Cu composites (a TC of 244.5 W/(m·K)). Zhou et al. [27] reported that the TC of GFs/Al composites with an average size of 500 μm reached 400 W/(m·K) with 50 vol% of GFs, while GFs/Al composites with an average size of 100 μm were about 320 W/(m·K). As the content of GFs with an average size of 500 μm reached 60%, the TC of GFs/Al composites reached 530 W/(m·K). Li et al. [28] provided a process to prepare highly aligned GFs/Al composites with different sizes of GFs by squeeze casting. As the average size of GFs increased from 150 μm to 495 μm, the in-plane TC increased from 548 to 664 W/(m·K). The TC of GFs/Al composites amazingly reached 714 W/(m·K), as the average size of GFs was 495 μm and volume fraction was 70%. Highly aligned GFs/Al composites exhibited great potential for heat dissipation. However, the fabrication process of the composites is very complicated for industrial production.

In this work, we focus on the preparation of GFs/Cu composites with ultrahigh TC. Considering the anisotropic TC of GFs, novel strategies for controlling the alignment of GFs in the copper matrix are successfully developed. Copper foil is firstly introduced into GFs/Cu composites, replacing the easily oxidized Cu powders. Not only is copper foil in the metal matrix, but also the basis of aligned GFs. The in-plane TC of GFs/Cu composites increases from 503 to 741 W/(m·K), as the volume fraction of GFs is 30~70%. The alignment of GFs, microstructures, interfaces and TC modeling are thoroughly discussed. The fabrication process with our approach is scalable and is easy used in industry. The advanced GFs/Cu composites can be realized to help cool modern electronics in the near future.

## 2. Materials and Methods

### 2.1. Materials and Reagents

Commercial graphite flakes (GFs) with an average in-plane size of 500 μm and thickness of 50 μm were supplied by Qingdao haida Graphite Co., Ltd. (Qingdao, China). The copper foil with a purity of >99.9%, having a series size of 6, 15, 50 and 100 μm, were provided by Jiangxi Copper Co., Ltd. (Nanchang, China). The polyvinyl butyral (PVB) with a purity of >99.9% was purchased from Sinopharm Chemical Reagent Co., Ltd. (Shanghai, China). The other reagents used in the electroless plating and tape casting process are commercial products.

### 2.2. Cu Coating Grown on the Surface of GFs

In order to improve the interfacial strength between GFs and Cu, a Cu coating was grown on the surface of the GFs by electroless plating method [25,29]. The main steps of electroless plating procedures are introduced as follows: (i) The GFs are put into 600 mL/L H_2_SO_4_ solution and stirred in the water bath with a speed of 80 r/min at 60 °C for 60 min; (ii) The GFs are treated by an ultrasonic instrument with the mixture of 20 g/L SnCl_2_ solution and 60 mL/L HCl solution for 20 min at room temperature; (iii) Continuing ultrasonic process with the mixture of 0.3 g/L PdCl_2_ solution and 5 mL/L HCl solution for 20 min in room temperature; (iv) The pretreated GFs are put into the mixture solution of 50 g/L CuSO_4_, 15 g/L Zn and 10 mL/L and stirred in the water bath at a speed of 150 r/min at 40 °C for 60 min. The GFs were cleaned with pure water at the end of each step above. (v) The treated GFs were dried by the electric vacuum drying oven at 80 °C.

### 2.3. Tape Casting Process

The GFs were well arranged on the surface of copper foil by tape casting process. The PVB– ethanol solution (the mass ratio of PVB and ethanol absolute was 1:7) was pre-mixed using a mixer at a speed of 150 r/min for 10 min. Then Cu coated GFs were put into the PVB–ethanol solution mixed at a speed of 200 r/min for 20 min and we then obtained the slurry for the tape casting process. The slurry was cast on the surface of 6 μm copper foil with a limited height of 60 μm controlled by a blade and pressed by a rubber roller. After that, it was dried at 80 °C for 5 min, and we obtained a special GFs–Cu layer that aligned Cu coated GFs coated on copper foil. The volume fraction of GFs in a single GFs–Cu layer was about 90% in reality.

### 2.4. Preparation of GFs/Cu Composites

The graphite flakes/copper foil (GFs/Cu) composites were prepared by hot-pressing process with the volume fraction of GFs ranging from 30% to 70%. To obtain the GFs/Cu composites with different content of GFs, the extra copper foil was put between two GFs–Cu layers (e.g., one layer of 100 μm copper foil for 30 vol% GFs/Cu composites, three layers of 15 μm copper foil for 30 vol% GFs/Cu composites and two layers of 6 μm copper foil for 70 vol% GFs/Cu composites). A number of GFs–Cu layers and copper foil were stacked into a graphite mold in turn with a layer-by-layer structure and sintered in the vacuum hot press furnace at 1040 °C and 40 MPa for 2 h [24,26]. The volume ratio was mainly controlled by adding different copper foil between two GFs–Cu layers and the calculation deviation of GFs volume fraction was 5%. The complete preparation process of aligned graphite/Cu composites was showed in Figure 1.

### 2.5. Material Characterization

The phase analysis of materials and obtained samples were characterized by X-ray diffraction (XRD, D8 ADVANCE DAVINCI, Karlsruhe, Germany) with monochromatic Cu-Kα radiation. The thickness of the Cu coating on the surface of the GFs was characterized by focused ion beam (FIB, Auriga, Oberkochen, Germany). Morphology of materials and samples were observed by scanning electron microscope (SEM, FEI Quanta FEG 250, Hillsboro, OR, USA) employing accelerating voltages of 20 kV, and the element distribution across the interface of GFs/Cu composites was analyzed by energy disperse spectroscopy (EDS). The thermal diffusivity (*α*) of composites was measured with laser flash apparatus (LFA467, Selb, Germany) at 25 °C. The density (*ρ*) of composites was measured with solid densimeter (AKR-220SD, Shenzhen, China) at room temperature. The specific heat (*C_p_*) was characterized by differential scanning calorimetry (DSC 214, Selb, Germany) at 25 °C. The values of thermal conductivity (K) were calculated by the equation:(1)$$K=\alpha \times \rho \times C_{p}$$

## 3. Results and Discussion

### 3.1. Microstructure of Materials and Composites

Figure 2a,b shows the morphologies of the raw GFs and Cu coated GFs, respectively. It is obvious to observe that a well coated layer was grown on the whole surface of the GFs. In order to get more information of the coating layer, the surface of coated graphite was peeled by FIB technique, as shown in Figure 2c. It could be found that a thickness of 1.8 μm coating layer was covered on the surface of GFs. The SEM image of the coated GFs–Cu foil layer was shown in Figure 2d. The coated GFs were well oriented on the surface of the Cu foil and partly contacted with others.

To further determine the composition of the coating on the surface of the GFs, the XRD patterns of raw GFs and coated GFs were obtained and are shown in Figure 3, with a two heat angle scanning range of 20°~80° and a scan speed of 4°/min. The (002) and (004) crystal planes of the graphite phase are indexed in Figure 3a,b. In addition, it can be found that the other two peaks of (111) and (200) crystal planes of Cu exist in Figure 3b, which proves that the primary element of the coating layer is Cu. The relative intensity of the Cu phase peaks is a little weak, because the Cu coating occupies low mass ratio of coated GFs. The in-plane and through-plane XRD patterns of the coated GFs/Cu foil composites are shown in Figure 3c,d, respectively. No other phases except Cu and graphite have been indexed, which indicates that the PVB–ethanol solution was well broken up or disappeared from the composites after the vacuum hot press process. The high relative intensity of graphite (002) crystal plane was indexed as the consequence of high orientation of GFs in Figure 3c.

Figure 4a,b shows the typical side-view (in Z plane, X–Z or Y–Z plane) and top-view (in X–Y plane) morphology of 70 vol% GFs/Cu composite. The dark region represents GFs while the light region represents the Cu matrix. As is shown in Figure 2d, the GFs were covering the surface of others in a certain area before the hot pressing process, which provided the possibility of long size continuous GFs. The size of the continuous dark region is much more than 500 μm, which is the average size of GFs in Figure 4a. The edges of GFs partly coincided with surroundings in Figure 4b, corresponding to Figure 2d. All these indicate that the single GF had connected with others to a certain degree, which is beneficial for forming high TC channels and improves the TC of the composite in the X–Y plane. In order to explore the forms of continuous GFs, the high-magnification of corresponding regions in Figure 4a were provided in Figure 4a1–a3. Some GFs contacted with others by edges, which may cause bending deformation in Figure 4a1. Some GFs were clamped by pieces of GFs in Z plane direction and connected edges by the other GFs in the X–Y plane direction without edge deformation in Figure 4a2. The other GFs overlapped with each other, with about half of the surface area in Figure 4a3. Those connection ways contribute to the forming of a highly thermal network of GFs/Cu composites in the basis of the GFs. It also can be found that the GFs network was well packaged by Cu matrix in Figure 4a,b. It might be that the Cu foil was in a semi-melted state at 1040 °C and the semi-melted Cu filled in the pores of the GFs network under 40 MPa pressure. Figure 4c shows the interfacial micro-structure and EDS line-scan analysis of GFs and Cu matrix. The element analysis showed a high diffraction peak of C and low diffraction peak of Cu in the graphite phase, as is reversed in the Cu matrix. There was an excessive boundary of C-Cu interface but no porosity or defects had been detected. Owing to the Cu coating on the surface of GFs, the GFs were not directly connected with each other. Cu coating on the surface of GFs provided transition area for the combination of GFs–Cu and GFs-GFs.

### 3.2. Quantitative Analysis of GFs Alignment

The alignment of GFs is significantly related to the thermal performance of composites because of its extremely anisotropic TC data. <*cos*^2^*β*> [30] is used to align the degree of GFs, given by
(2)$$<cos^{2}\beta >=\frac{\int \rho \left(\beta \right)cos^{2}\beta sin\beta d\beta }{\int \rho \left(\beta \right)sin\beta d\beta }$$
where *β* is an acute angle of GFs high TC plane direction to the perpendicular direction of hot pressing axis, which is marked in Figure 5a. The *ρ(β)* is a statistical function used to describe the alignment degree of GFs in composites. When <*cos*^2^*β*> = 1/3, the GFs alignment is close to random distribution. <*cos*^2^*β*> = 0 or <*cos*^2^*β*> = 1 shows that the GFs alignment is absolutely parallel to hot-pressing axis or perpendicular to hot-pressing axis, respectively [31].

Figure 5a–c shows the SEM images of GFs/Cu composites in the Z plane as the volume fraction of GFs being 30%, 50%, 70%, respectively. Most of the GFs are well arranged in the composites with a consistent orientation. It can be found that there exists a highly paralleled structure where the GFs layer alternates with Cu layer in Figure 5a. As the volume fraction of GFs increased from 30% to 70%, the content of Cu layers was decreased and GFs layers network gradually appeared in Figure 5b,c. Figure 5d–f shows the frequency of *β* distribution in different angel range and the fitted function *ρ(**β**).* The *β* and *ρ(**β**)* were obtained by the procedure as follows. At first, the length(*L*) and angel(*β*) of single GF in SEM images were replaced by black lines with similar data. Then we got *L* and *β* data information by calculating these black lines. Finally, the *ρ(β)* function was fitted with the obtained *β* data by the ExpDec1 function derived from nonlinear curve fit procedure, given by
(3)$$\rho \left(\beta \right)=A_{1}e^{\left(-\beta /t_{1}\right)}+y_{0}$$
where *A*_1_, *t*_1_, *y*_0_ are fitting parameters. The <*cos*^2^*β*> values of GFs/Cu composites with different GFs volume fraction can be obtained. Figure 5d–f and Table 1 show the frequency analysis of *β* and function fitting of *ρ*(*β*). Most of *β* concentrates on 0~10° and more than 90% of *β* are less than 20°, further confirming the highly aligned GFs. Though the <*cos*^2^*β*> values decreased from 0.97 to 0.87 as the volume fraction of GFs increased from 30% to 70%, they are much larger than 1/3.

### 3.3. Thermal Properties and Modeling

Table 1 depicts the thermal properties of GFs/Cu composites with different contents of GFs. The density apparently decreases with the increase of GFs content. The thermal diffusion coefficient in X–Y plane sharply increases from 175 to 349 mm^2^/s as the volume fraction of GFs increases from 30% to 70%, and the thermal diffusion coefficient in Z plane decreases from 33 to 20 mm^2^/s gently in reverse. According to Equation (1), the TC values of GFs/Cu composites are calculated in Table 1. The highly aligned GFs have an apparent effect on thermal conductivity of GFs/Cu composites. It can be seen that the TC is extremely high in X–Y plane, while relatively low in Z plane, which indicates the anisotropic thermal properties of GFs/Cu composites. As with 70 vol% GFs, the TC of composites in X–Y plane and Z plane are 741 and 42 W/(m·K), respectively.

Herein, the *V_g_* is the volume fraction of GFs, *ρ*, *C_p_*, *α*, TC are the density, specific heat capacity, thermal diffusion coefficient and thermal conductivity of the GFs/Cu composites, <*cos*^2^*β*> is the alignment degree of GFs in the GFs/Cu composites.

The TC of GFs/Cu composite can be predicted by layers-in- parallel model (i.e., parallel to the X–Y plane) and layers-in-series model (i.e., perpendicular to the X–Y plane) with the ideal orientation of GFs. The model is given by [32]
(4)$$K_{C}^{L}{=K}_{g}^{L}V_{g}+K_{m}\left(1-V_{g}\right)$$
(5)$$\frac{1}{K_{C}^{T}}=\frac{V_{g}}{K_{g}^{T}}+\frac{1-V_{g}}{K_{m}}$$
where *K_C_*, *K_g_*, *K_m_* are the TC of composite, graphite and Cu matrix, respectively. The subscripts ‘*L*’, ‘*T*’ mean the direction parallel to the X–Y plane and perpendicular to the X–Y plane, respectively. Taking the interfacial thermal resistance of Cu and GFs into account, the analytical TC of reinforcement can be replaced by an “effective” TC, $K^{eff}$, to improve the reliability of analysis. Considering the anisotropic TC of GFs, $K_{L}^{eff}$ in X–Y plane direction and $K_{T}^{eff}$ in Z plane direction are given by [32]
(6)$$K_{L}^{eff}=\frac{K_{g}^{L}}{1+\frac{2K_{g}^{L}}{hD}}$$
(7)$$K_{T}^{eff}=\frac{K_{g}^{T}}{1+\frac{2K_{g}^{T}}{ht}}$$
where *K_L_*, *K_T_* are the TC of GFs in in-plane direction and through-plane direction. *D* is the diameter of the GFs and t is the thickness of GFs. The average *D* = 500 μm, t = 50 μm. *h* is the interfacial thermal conductance. It can be calculated by the acoustic mismatch model (AMM), given by [33]
(8)$$h≅\frac{1}{2}\rho_{m}C_{m}\frac{v_{m}^{3}}{v_{i}^{2}}\frac{\rho_{m}v_{m}\rho_{g}v_{g}}{{\left(\rho_{m}v_{m}+\rho_{g}v_{g}\right)}^{2}}$$
where *ρ*, *C*, *v* are the density, the specific heat capacity and phonon velocity, respectively. The subscripts of ‘*m*’, ‘*g*’ represent Cu matrix and graphite in this article, respectively. The material parameters for calculating *h* are given in Table 2 and we obtained $K_{L}^{eff}$ = 880 W/(m·K), $K_{T}^{eff}$ = 36 W/(m·K), *h* = 2.93 × 10^7^ W/(m^2^·K).

Considering the GFs alignment and interfacial thermal resistance, the anisotropic TC data of GFs/Cu composite are deeply investigated by the effective medium approximation (EMA) model [30] with laminated flat plate reinforcement. Similar to aligned graphene composites [31], the calculation of TC of composites in X–Y plane can be replaced by:(9)$$K_{C}^{L}{=K}_{m}\left\{\frac{2+V_{g}\gamma_{L}\left(1{\;+\;<cos}^{2}\beta >\right)}{2-V_{g}\gamma_{T}\left(1\;-\;{<cos}^{2}\beta >\right)}\right\}$$
with
(10)$$\gamma_{L}=\frac{K_{L}^{eff}-K_{m}}{K_{m}}$$
(11)$$\gamma_{T}=\frac{K_{L}^{eff}-K_{m}}{K_{L}^{eff}}$$

We obtained *γ_L_* = 1.316, *γ_T_* = 0.568 and $K_{C}^{L}$ can be calculated. The values of TC in Z plane can be calculated by Equations (5) and (7). The TC model of GFs/Cu composites is shown in Figure 6.

Figure 6 shows the comparison on TC of different GFs/Cu composites. The raw GFs/Cu composites, B_4_C coated GFs/Cu composites [23] and SiC coated GFs/Cu composites [24] were fabricated by powder metallurgy method in an vacuum hot pressing furnace, with the GFs contents of 40~70 vol%. As shown in Figure 6a, the B_4_C coated GFs/Cu composites and SiC coated GFs/Cu composites exhibit lower TC than that of GFs/Cu composites. The TC of B_4_C coated GFs/Cu composites and SiC coated GFs/Cu composites are 449~608 W/(m·K) and 460~610 W/(m·K), respectively, while GFs/Cu composites have 477~676 W/(m·K). It indicates that even if coatings enhanced the interfacial binding strength of GFs and Cu, it may not help to increase the TC of composites, for the interface thermal resistance of ‘GFs-coatings-Cu’ is higher than ‘GFs–Cu’. Therefore, Cu coating was introduced in this work by electroless plating to keep the balance of interfacial binding strength and interface thermal resistance in the GFs/Cu composites. As a result, the TC of aligned GFs/Cu composites (our work) reaches 503~683 W/(m·K) with the volume fraction of 30~60%, which is slightly higher than GFs/Cu composites (Reference 23) with GFs content of 40~70%. When the volume fraction of GFs increases to 70%, the aligned GFs/Cu composites exhibits a high TC value of 741 W/(m·K). In Figure 6b, those different GFs/Cu composites have a close thermal performance in Z plane. The TC values of raw GFs/Cu composites, B_4_C coated GFs/Cu composites, SiC coated GFs/Cu composites and our work are 74~40, 71~40, 77~36 and 72~42 W/(m·K), respectively, with GFs content from 40 to 70 vol%. This indicates that interfacial problems and alignment of GFs have a limiting influence on the Z-plane TC of GFs/Cu composites.

The anisotropic thermal properties of GFs (with in-plane TC of 1000 W/(m·K) and through-plane TC of 38 W/(m·K)) causes the anisotropic TC of GFs/Cu composites. The value of anisotropy (VA) can be used to evaluate the alignment degree of GFs in composites, which is the ratio of TC in parallel to high thermal transfer direction (X–Y plane) and perpendicular to high thermal transfer direction (Z plane). As the volume fraction of GFs increases up to 70%, the VA of GFs/Cu composites and our high aligned GFs/Cu composites is 16.90 and 17.64, respectively. The result confirms the superiority of our work in controlling the alignment of GFs in composites.

## 4. Conclusions

In conclusion, ultrahigh thermal conductivity of aligned graphite flakes/Cu foil composites were fabricated by vacuum hot pressing process. The thermal conductivity of highly aligned graphite flakes/copper composites exhibited extremely anisotropy in the in-plane and through-plane. As the volume fraction of graphite flakes increased from 30% to 70%, the in-plane thermal conductivity was ultrahigh and increased from 503 to 741 W/(m·K), while it decreased in the through-plane, which decreased from 95 to 42 W/(m·K). The alignment of graphite flakes and interfacial thermal resistance were taken into account to modify the thermal conductivity model for graphite/copper composites. The fabrication and modeling also suit other graphite flakes/metal matrix composites. The advanced composites can be realized to help cool modern electronics in the near future.

## Figures and Tables

**Figure 1 materials-13-00046-f001:**
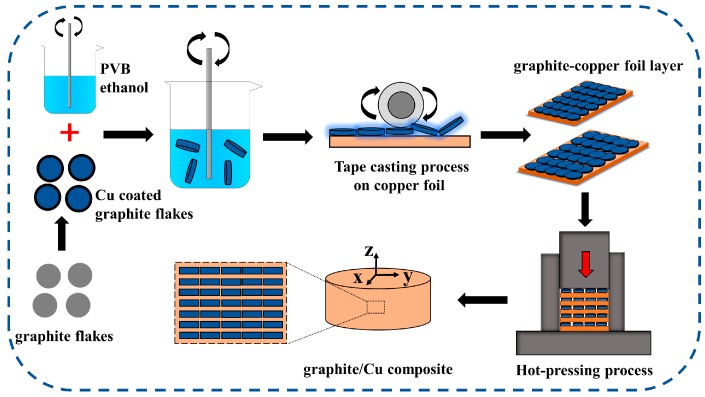
Schematic illustration of the preparation process of aligned graphite/Cu composites.

**Figure 2 materials-13-00046-f002:**
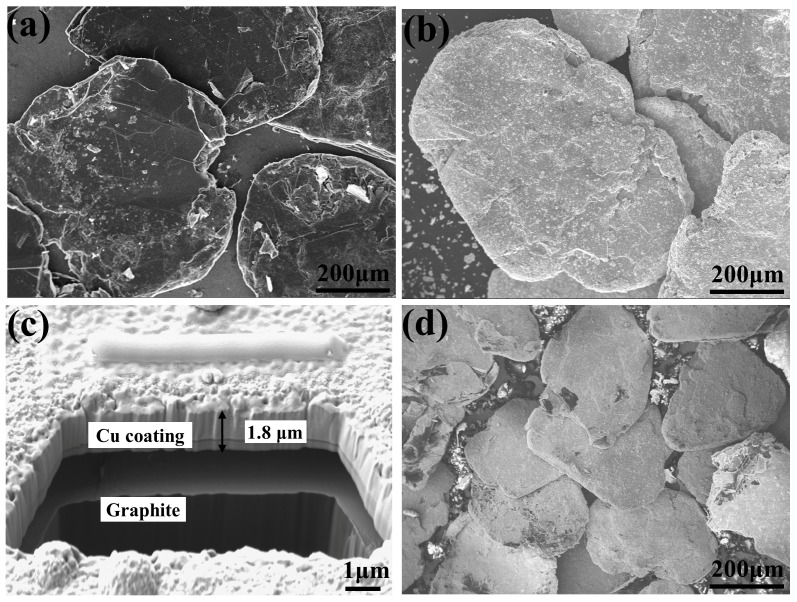
SEM images of (**a**) raw graphite flakes (GFs), (**b**) coated GFs, (**c**) coating on the surface of the GFs (**d**) coated GFs–Cu layer.

**Figure 3 materials-13-00046-f003:**
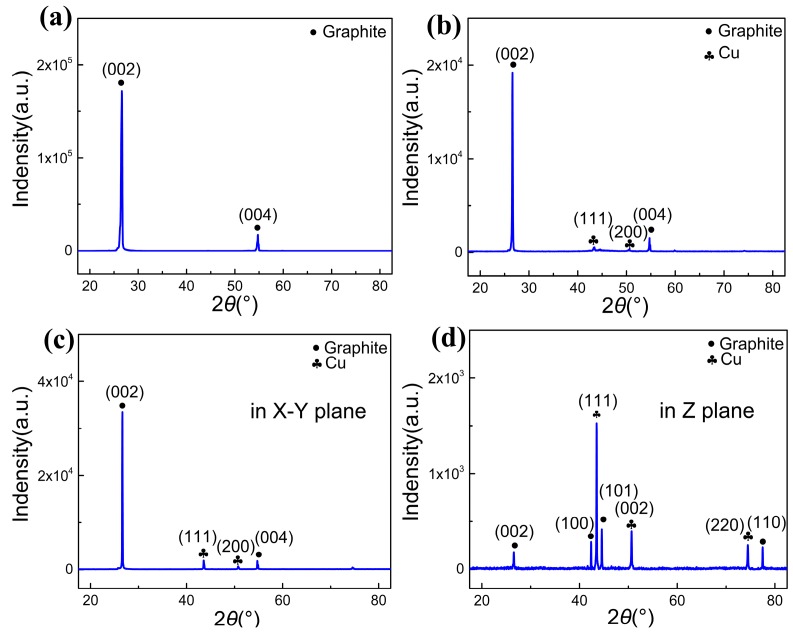
XRD patterns of (**a**) raw GFs, (**b**) coated GFs, (**c**) coated GFs/Cu composites in X–Y plane, and (**d**) coated GFs/Cu composites in Z plane.

**Figure 4 materials-13-00046-f004:**
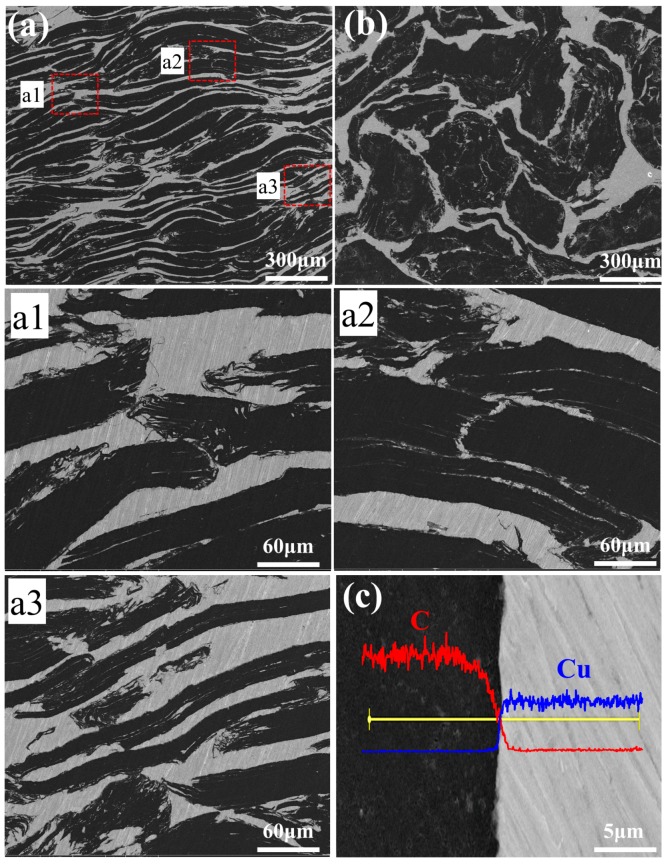
Morphology of GFs/Cu composites (**a**) in Z plane (**a1**, **a2**, **a3** are the 5X magnification of corresponding regions in (**a**), (**b**) in X–Y plane; (**c**) interfacial micro-structure and EDS line-scan analysis.

**Figure 5 materials-13-00046-f005:**
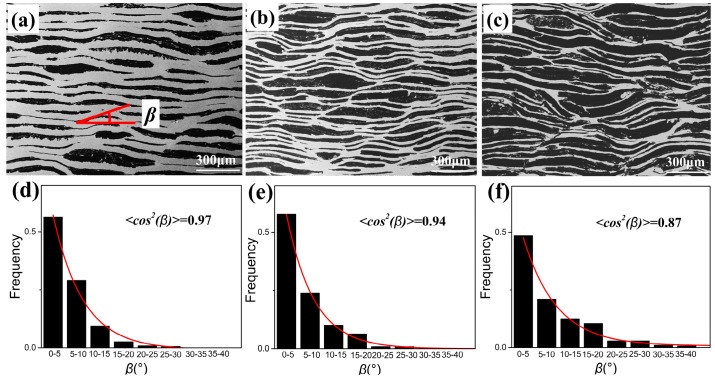
(**a**–**c**) SEM images of GFs/Cu composites in Z plane as the volume fraction of GFs is 30%, 50% and 70%, respectively; (**d**–**f**) Corresponding frequency analysis of *β* and function fitting of *ρ(**β).*

**Figure 6 materials-13-00046-f006:**
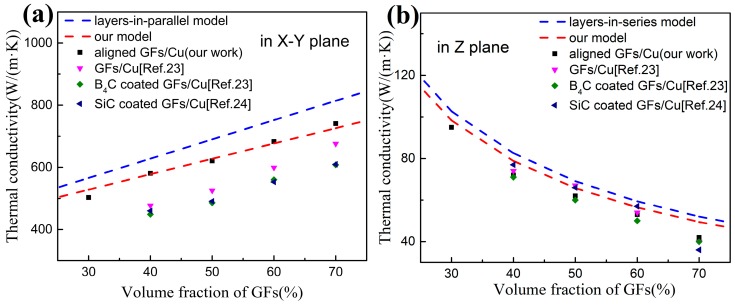
The thermal conductivity of GFs/Cu composites (**a**) in X–Y plane, (**b**) in Z plane.

**Table 1 materials-13-00046-t001:** Properties (GFs content and alignment degree, density, specific heat capacity thermal diffusion coefficient and thermal conductivity) of GFs/Cu composites.

*V_g_* (%)	*ρ* (g/cm^3^)	*C_p_* (J/(g·K))	*α*(mm^2^/s)		TC(W/(m·K))		<*cos*^2^*β*>
			X–Y	Z	X–Y	Z	
30	6.831	0.421	175	33	503	95	0.97
40	6.142	436	217	27	581	72	0.96
50	5.492	455	249	25	621	62	0.94
60	4.825	478	296	23	683	53	0.91
70	4.174	509	349	20	741	42	0.87

**Table 2 materials-13-00046-t002:** Material parameters for theoretical calculation.

Material	Density (g/cm^3^)	Specific Heat Capacity (J/(g·K))	Phonon Velocity (m/s)	Thermal Conductivity (W/(m·K))	References
Graphite	2.260	0.710	14800	1000^X–Y^	[34]
				38^Z^	
Cu	8.900	0.385	2500	380	[35]

The TC value of Cu is measured by LFA467.

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
