# Peer review of "High Thermal Conductivity and Anisotropy Values of Aligned Graphite Flakes/Copper Foil Composites"

_materials, 2019, doi:10.3390/ma13010046_

Round 1

Reviewer 1 Report

The subject of research is very interesting. The article is also interesting. However, there are many typographical errors in the article, for example “Gemany” instead of Germany, “indensity” instead of intensity… Typographical errors are also in the figures. The spelling should be checked carefully.

Chapter 2.3 Tape casting process should be better written. Does PVB mean polyvinyl butyral? This abbreviation should be explained in the text.

In line 89 there is information that 6 mm thick foil was used. Earlier (line 70) it was written that the thickness of the foil was from 6 to 1000 µm. This requires clarification.

Author Response

Question 1. The subject of research is very interesting. The article is also interesting. However, there are many typographical errors in the article, for example “Gemany” instead of Germany, “indensity” instead of intensity… Typographical errors are also in the figures. The spelling should be checked carefully.

Reply: Thank you for your comments. Typographical errors have been corrected and marked in red color in the revised manuscript.The article are also checked by my native English-speaking colleague.

Question 2. Chapter 2.3 Tape casting process should be better written. Does PVB mean polyvinyl butyral? This abbreviation should be explained in the text.

Reply: Thank you for your comments. The PVB does mean polyvinyl butyral and the full name for the PVB abbreviation has been added in line 96 and marked in red color in revised version.

Question 3. In line 89 there is information that 6 mm thick foil was used. Earlier (line 70) it was written that the thickness of the foil was from 6 to 1000 µm. This requires clarification.

Reply: Sorry for our mistake. The “6 mm” is changed to “6 µm” in the revised version.

Reviewer 2 Report

In this form I cannot suggest accepting this article and therefore authors must major repaired it. The article can be interesting and helpful, but it has several mistakes and bad formalities (above). Also English writing should be improved.

Author Response

Thank you for your suggestion.We have provided the detailed explaination for your questions in the attached files.

Reviewer 3 Report

Comments:

Why there is highly anisotropy in the in the thermal conductivity of GFs/Cu composites? Is this anisotropy is good? If not good, should there be an optimum condition for alignment and anisotropy? In the introduction part the author showed that carbon based material are very good for heat dissipation. Then why there is need of carbon based composites? The should show some recent work on the alignment of the Gfs/Cu, GFs/Al etc, so that it become clear to know the worth of this work. Need to review” a Cu layer was coated on GFs by 75 electro-less plating method” Is it Cu layer on Fs or GFs layer on Cu? What are the main laboratory and commercial achievement of this wor. Please mention in the conclusion and introduction part. Use x103 Instead of 1000..etc in Fig.3 and etc.

Author Response

(The authors gave the same response as above.)

Round 2

Reviewer 2 Report

The Authors correct repaired my comments and several mistakes and now I can agree with the article for publishing in journal.